# High-Throughput Sequencing Reveals That *Rotundine* Inhibits Colorectal Cancer by Regulating Prognosis-Related Genes

**DOI:** 10.3390/jpm13030550

**Published:** 2023-03-20

**Authors:** Lingyu Huang, Tongxiang Zou, Wenken Liang, Chune Mo, Jianfen Wei, Yecheng Deng, Minglin Ou

**Affiliations:** 1Key Laboratory of Ecology of Rare and Endangered Species and Environmental Protection, Ministry of Education of China/College of Life Science, Guangxi Normal University, Guilin 541006, China huanglingyu99@163.com (L.H.); zoutongx@126.com (T.Z.); wenken199510@163.com (W.L.); 15560159075@163.com (J.W.); 2Central Laboratory, Guangxi Health Commission Key Laboratory of Glucose and Lipid Metabolism Disorders, The Second Affiliated Hospital of Guilin Medical University, Guilin 541000, China; cemo@glmc.edu.cn

**Keywords:** *rotundine*, high-throughput sequencing, colorectal cancer, prognosis-related genes

## Abstract

Background: *Rotundine* is an herbal medicine with anti-cancer effects. However, little is known about the anti-cancer effect of *rotundine* on colorectal cancer. Therefore, our study aimed to investigate the specific molecular mechanism of *rotundine* inhibition of colorectal cancer. Methods: MTT and cell scratch assay were performed to investigate the effects of *rotundine* on the viability, migration, and invasion ability of SW480 cells. Changes in cell apoptosis were analyzed by flow cytometry. DEGs were detected by high-throughput sequencing after the action of *rotundine* on SW480 cells, and the DEGs were subjected to function enrichment analysis. Bioinformatics analyses were performed to screen out prognosis-related DEGs of COAD. Followed by enrichment analysis of prognosis-related DEGs. Furthermore, prognostic models were constructed, including ROC analysis, risk curve analysis, PCA and t-SNE, Nomo analysis, and Kaplan–Meier prognostic analysis. Results: In this study, we showed that r*otundine* concentrations of 50 μM, 100 μM, 150 μM, and 200 μM inhibited the proliferation, migration, and invasion of SW480 cells in a time- and concentration-dependent manner. *Rotundine* does not induce SW480 cell apoptosis. Compared to the control group, high-throughput results showed that there were 385 DEGs in the SW480 group. And DEGs were associated with the Hippo signaling pathway. In addition, 16 of the DEGs were significantly associated with poorer prognosis in COAD, with *MEF2B*, *CCDC187*, *PSD2*, *RGS16*, *PLXDC1*, *HELB*, *ASIC3*, *PLCH2*, *IGF2BP3*, *CLHC1*, *DNHD1*, *SACS*, *H1-4*, *ANKRD36*, and *ZNF117* being highly expressed in COAD and *ARV1* being lowly expressed. Prognosis-related DEGs were mainly enriched in cancer-related pathways and biological functions, such as inositol phosphate metabolism, enterobactin transmembrane transporter activity, and enterobactin transport. Prognostic modeling also showed that these 16 DEGs could be used as predictors of overall survival prognosis in COAD patients. Conclusions: *Rotundine* inhibits the development and progression of colorectal cancer by regulating the expression of these prognosis-related genes. Our findings could further provide new directions for the treatment of colorectal cancer.

## 1. Background

According to the International Agency for Research on Cancer, global colorectal cancer incidence will increase by 63% and mortality to 73% by 2024 compared to the 2020 Global Cancer Statistics report [1]. New advances have been made in recent years in the search for new treatments for COAD; however, the incidence of COAD is still on the rise year by year. The current clinical treatment of colorectal cancer is mainly through surgical resection, post-surgical adjuvant therapy, and chemotherapy. However, recurrence and metastasis of surgical treatment is one of the causes of death in colorectal cancer patients. Chemotherapy drugs include western drugs such as 5-fluorouracil, oxaliplatin, capecitabine, and irinotecan [2,3]. However, side effects from chemotherapy are inevitable, for example, nausea, vomiting, diarrhea, and peripheral neurotoxicity. Finding therapeutic targets based on high-throughput sequencing and bioinformatics analysis is gaining importance in the treatment protocols for COAD [4]. The application of advanced high-throughput mRNA sequencing technologies has enabled researchers to fully explore the pathogenesis of cancer.

*Rotundine* (molecular formula: C_21_H_25_NO_4_), also known as *L-tetrahydropalmatine*(*L-THP*), is an alkaloid extracted mainly from the active ingredients of *Stephania epigaea Lo* and *Corydalis yanhusuo*. It is included in the Chinese Pharmacopoeia. *Rotundine* has been reported to be widely used for its analgesic and hypnotic effects [5,6], and its pharmacological effects include anti-inflammatory, anti-virus, and anti-tumor [7]. For example, *L-THP* inhibits the growth of hepatocellular carcinoma by promoting HepG2 cell autophagy through activation of the AMPK pathway [8]. The majority of breast cancer (BCa) patients were diagnosed as estrogen receptor alpha (ERα) positive. A study by Xiaohong Xia et al. found that *L-THP* achieved the effect of inhibiting BCa proliferation by blocking the cell cycle and promoting ERα degradation [9]. And *Rotundine* has been shown to have good results in clinical trials for the treatment of addiction [10,11]. Although *rotundine* has been studied for its anti-cancer effect. However, the mechanism of action on COAD has not been reported systematically; therefore, it becomes urgent to investigate the mechanism of action of *rotundine* on COAD.

We learned that the herbal extract *rotundine* has cancer-inhibitory effects. Therefore, in this study, we used MTT assay and cell scratch assay to detect the effects of *rotundine* on the proliferation, invasion, and migration of colorectal cancer cells SW480. Further, high-throughput sequencing was used to explore the molecular mechanism of *rotundine* inhibition in SW480 cells in depth. Finally, bioinformatics analysis was performed in conjunction with TCGA public database to validate the aberrantly expressed mRNAs. These aberrantly expressed mRNAs may serve as potential biomarkers for COAD treatment.

## 2. Materials and methods

### 2.1. Materials

*Rotundine* was purchased from Chengdu Desite Biotechnology (Chengdu, China) with CAS number 10097-84-4 and a purity equal to 99.77%. Following the principle that a DMSO concentration of less than 0.1% is non-toxic to cells, we first dissolved the *rotundine* powder with DMSO and then added PBS to configure the concentration to 10 mM for storage in –80 °C refrigerator. Before performing the experiments, the 10 mM *rotundine* solution was diluted into 50 μM, 100 μM, 150 μM, and 200 μM *rotundine* solutions using RPMI-1640 medium with 10% FBS and 1% penicillin-streptomycin solution.

### 2.2. Cell Culture

SW480, a human colorectal cancer cell line, was cultured in RPMI-1640 medium with 10% FBS and 1% penicillin–streptomycin solution. Finally, it was placed in a humidified incubator at 37 °C and 5% CO_2_.

### 2.3. MTT Assay

In a 96-well plate, 5 × 10^4^ cells were added and incubated in the incubator for 24 h. After the cells were plastered, RPMI-1640 culture medium and *rotundine* solution were added, respectively. Three replicate wells were made for each group. When the incubation time is reached, add 20 μL MTT solution, shake well, and continue to incubate for 4 h. Carefully aspirate and discard the supernatant, add 150 μL DMSO and place in an enzyme marker with a low-speed shaking for 10 min. Finally, measure the OD value at 490 nm.

### 2.4. Cell Scratch Assay

In a 6-well plate, 6 × 10^5^ cells were added and incubated in the incubator for 24 h. Draw a straight line perpendicular to the center of the culture wells with a sterile 200 µL pipette tip. Discard the old medium and wash twice with 2 mL of PBS. The control group was added with 3 mL of RPMI-1640 complete medium. To the drug group, add 3 mL each of 50 μM, 100 μM, 150 μM, and 200 μM *rotundine* base solutions. Scratch repair was observed after every 24 h.

### 2.5. Apoptosis Assay

Apoptosis of *rotundine*-treated SW480 was measured with FITC Annexin V Apoptosis Detection Kit I. *Rotundine* (0, 50, 100, 150, 200 μM) was applied to SW480 cells for 72 h, and then stained by Annexin V-FITC/PI for 15 min and analyzed by BF 700 flow cytometer.

### 2.6. High-Throughput mRNA Sequencing

We divided the experiment into experimental and control groups; the experimental group was the intervention culture of SW480 cells with 50 μM, 100 μM, 150 μM, and 200 μM *rotundine*, and the control group was the conventional culture with RPMI-1640. The samples of both groups were then subjected to high-throughput mRNA sequencing according to the protocol provided by the manufacturer. First, RNA was extracted from the samples and mRNA was enriched using Oligo(dT) beads. Next, mRNA was interrupted by adding reagents, then cDNA was synthesized by reverse transcription, end repair, end addition of “A”, and splice ligation were performed on cDNA, and finally, cDNA was amplified by PCR to construct a cDNA library by PCR. Finally, mRNA sequencing is performed by combined probe-anchored polymerization (cPAS).

### 2.7. Identification of Differentially Expressed Genes

The cutoff values for screening differentially expressed genes (DEGs) were based on |log_2_FC| ≥ 1 and *p* < 0.05. DEGs were divided into high and low-expression groups based on log_2_FC values. Volcano plots represent the distribution results of the two groups. A heat map was used to represent the gene expression profiles of DEGs.

### 2.8. Enrichment Analysis and Construction of Protein-Protein Interaction(PPI) Network

To explore the relevant signaling pathways and functions of DEGs. We performed KEGG and GO enrichment analyses of DEGs online using Dr. Tom system. Among them, GO enrichment analysis includes biological processes (BP), molecular functions (MF), and cellular composition (CC). In addition, we construct PPI networks to explore the potential connections between DEGs.

### 2.9. Bioinformatics Analysis Based on TCGA Database

We downloaded 41 normal tissues and 473 COAD tissues from the TCGA public database. Based on the COAD data, we used R language to explore the expression of differentially expressed genes in normal and cancer tissues and overall survival analysis. The aim was to explore the target genes of the inhibitory effect of *rotundine* on colorectal cancer. We also performed KEGG and GO enrichment analyses on these target genes. Finally, the target genes were then subjected to correlation analysis. The results were presented in the form of correlation network plots.

### 2.10. Prognostic Model Construction

We used the R package to model survival prognosis based on target genes. The risk scores were then calculated using the “dplyr” package, and the COAD patients in the TCGA dataset were divided into high-risk and low-risk groups. We evaluated the model by plotting risk curves and risk scatter plots, PCA, t-SNE, and Kaplan–Meier plotter using the “ggplot2′” package. In addition, we also plotted ROC curves and Nomo plots based on clinical characteristics and risk scores.

### 2.11. Statistical Analysis

All differences between the two groups were assessed using the student’s t-test. However, *p* < 0.05 was considered to be statistically significant.

## 3. Results

### 3.1. Effect of Rotundine on Cytotoxicity and Invasive Migration of SW480

MTT assay is widely used in detecting the cytotoxic activity of drugs on cells; therefore, we examined the toxicity of *rotundine* on SW480 cells by MTT assay. As shown in Figure 1a, *rotundine* was cytotoxic to SW480 at concentrations of 50 μM, 100 μM, 150 μM, and 200 μM. The proliferation of cells was significantly inhibited after *rotundine* action on SW480, and the inhibition rate was stronger with increasing *rotundine* concentration. The results of scratch experiments at 24 h, 48 h, and 72 h also showed that the healing of cell scratches was significantly slower after *rotundine* intervention in culture compared to the control group (Figure 1b).

### 3.2. Apoptosis Assay

To further investigate whether *rotundine* induced SW480 cell apoptosis, Annexin V-FITC/PI staining was performed after *rotundine* intervention on SW480 cells to observe the apoptosis. In our study, the rate of cell apoptosis was 15.78% in the control group and 9.38%, 10.81%, 7.59%, and 10.5% after the addition of *rotundine* (50, 100, 150, and 200 μM), respectively (Figure 2). According to these results, it was shown that the addition of *rotundine* intervention to SW480 culture did not induce cell apoptosis compared to the control group.

### 3.3. High-Throughput mRNA Sequencing Analysis

To further clarify the mechanism of *rotundine* regulation of COAD, we used high-throughput sequencing to analyze the mRNAs of the experimental and control group in detail. As shown in Figure 3a, there were 14,640 mRNAs expressed in the two groups. Among them, there were also 470 independent mRNAs in the experimental group and 790 independent mRNAs in the control group. mRNAs were visualized under the criteria of |log_2_FC| ≥ 1 and *p* < 0.05 (Table 1 and Appendix A). The results are presented as a volcano plot, where red dots indicate 85 up-regulated DEGs, green dots indicate 300 down-regulated DEGs, and gray dots indicate no-DEGs (Figure 3b). In addition, clustering heat map analysis revealed the expression profiles of 385 differentially expressed mRNAs between the experimental groups and control group (Figure 3c).

### 3.4. Enrichment Analysis and Construction of PPI Network

Gene annotation was performed on 385 DEGs. The first 20 analyses were visualized to elucidate the main biological functions of the DEGs. KEGG enrichment results showed that the main signaling pathways that occurred expression in the experimental and blank groups were the Hippo signaling pathway and intestinal immune network for IgA production (Figure 3d). Additionally, GO enrichment analysis included biological processes (BP), cellular components (CC), and molecular functions (MF). For example, functional enrichment of DEGs resulted in: negative regulation of the execution phase of apoptosis, plasma membrane, and receptor antagonist activity (Figure 3e–g). notably, a potential association of DEGs was found by constructing a PPI network (Figure 3h).

### 3.5. Gene Expression Analysis of 16 DEGs

As shown in Figure 4, by gene expression analysis, we found 16 DEGs were differentially expressed in normal and COAD tissues, namely myocyte enhancer factor 2B (*MEF2B*), coiled-coil domain containing 187 (*CCDC187*), pleckstrin and Sec7 domain containing 2 (*PSD2*), regulator of G protein signaling 16 (*RGS16*), plexin domain containing 1 (*PLXDC1*), ARV1 homolog (*ARV1*), DNA helicase B (*HELB*), acid-sensing ion channel subunit 3 (*ASIC3*), acid-sensing ion channel subunit 3 (*PLCH2*), insulin-like growth factor 2 mRNA binding protein 3 (*IGF2BP3*), clathrin heavy chain linker domain containing 1 (*CLHC1*), dynein heavy chain domain 1 (*DNHD1*), sacsin molecular chaperone (*SACS*), H1.4 linker histone (*H1-4*), ankyrin repeat domain 36 (*ANKRD36*), and zinc finger protein 117 (*ZNF117*). In addition, these 16 DEGs satisfied the following conditions: mRNA was down-regulated in SW480 cells after *rotundine* action compared with the control group, and the mRNA was up-regulated in COAD tissues compared with normal tissues. Conversely, after *rotundine* action on SW480, mRNA was up-regulated compared to the control group, and the mRNA was down-regulated in COAD tissue compared to normal tissue. Among them, *ARV1* expression was up-regulated after *rotundine* intervention, and the expression of the remaining 15 genes was down-regulated. Therefore, we judged these 16 genes as the targets of *rotundine* action to inhibit colorectal cancer.

### 3.6. Overall Survival Prognosis Analysis of 16 DEGs

The relationship between DEGs and survival prognosis was determined by comparing the overall survival rate of DEGs. As shown in Figure 5, the survival prognosis of highly expressed DEGs in COAD was poorer compared to normal tissues. Similarly, low-expressing *ARV1* had a poorer survival prognosis in COAD compared to normal tissue. The *p*-values for the above results were less than 0.05 between both groups, and all were statistically significant.

### 3.7. KEGG and GO Enrichment Analysis of 16 DEGs

KEGG enrichment results revealed 16 DEGs that may be involved in signaling pathways such as inositol phosphate metabolism, axon regeneration, and inflammatory mediator regulation of TRP channels (Figure 6a). Functional enrichment analysis using GO revealed that GO was significantly enriched in inner dynein arm, enterobactin transmembrane transporter activity, and enterobactin transport (Figure 6b–d).

### 3.8. Genetic Correlation Analysis of 16 DEGs

It was clearly observed by the correlation network plot and correlation heat map that the genes were closely and positively associated with each other (Figure 6e). Therefore, we suggest that *rotundine* can simultaneously down-regulate or up-regulate these 16 DEGs to inhibit the proliferative growth of COAD and improve the survival rate of patients.

### 3.9. Prognostic Model Analysis

Based on the median risk score, patients were divided into a high-risk group and a low-risk group (Figure 7a). Figure 6b shows the survival status and duration of survival of patients in relation to the risk score. As shown by Figure 7b, the number of deaths of patients in the high-risk group increased with increasing time. In addition, the results of PCA and t-SNE further indicate that the COAD cohort can be divided into high-risk and low-risk groups based on the median risk score (Figure 7c,d). The Kaplan–Meier survival analysis showed that patients in the high-risk group had a worse survival prognosis compared with the low-risk group, *p* < 0.0001 (Figure 7e). To demonstrate the accuracy of this prognostic model, we constructed ROC curves from 1–5 years (Figure 7f). Notably, the AUC values for 1–5 years predicted by our constructed model were all greater than 0.5, namely: 0.68, 0.72, 0.68, 0.71, and 0.72. In addition, to explore the risk assessment and survival probability of COAD patients in-depth, we combined risk scores and clinical factors to build Nomo plots (Figure 7g).

## 4. Discussion

Currently, the treatment of colorectal cancer is based on surgery and chemotherapy. However, many chemotherapy drugs have side effects such as nausea, vomiting, bone marrow suppression, peripheral neurotoxicity, and diarrhea [2,12]. Several studies have shown great progress in the study of bioactive compounds from natural plants in the treatment of tumors, both from animal model experiments and clinical trial studies [13,14]. Since *rotundine* relieves patients’ pain in clinical treatment, we believe that *rotundine* is non-toxic to normal cells or tissues [15,16,17,18]. It is worth noting that *rotundine* does not induce cells to undergo apoptosis. However, when combined with tamoxifen, *rotundine* can increase the sensitivity of BCa cells to tamoxifen to promote the apoptosis of death [9]. Therefore, it becomes necessary to search for natural plant active ingredients to treat colorectal cancer.

The MTT assay indirectly reflects cell survival and proliferation by detecting the number of viable cells [19]. In this study, we confirmed that *rotundine* is a natural plant active ingredients that can inhibit the proliferation and growth of colorectal cancer by MTT assay and cell scratching assay. Moreover, *rotundine* inhibited colorectal cancer in a time- and concentration-dependent manner. By flow cytometry, we found that *rotundine* did not induce apoptosis in colorectal cancer cells, similar to the study by Xia et al. who showed that *rotundine* did not induce apoptosis in breast cancer cells [9]. Therefore, we believe that *rotundine* exerts its anti-cancer effects by inhibiting the proliferation, migration, and invasion of colorectal cancer cells. Accordingly, the aim of this study was to investigate the specific molecular mechanisms by which *rotundine* inhibits colorectal cancer cells. By which genes *rotundine* exerts its inhibitory effect after intervention in colorectal cancer cells compared to cell samples without *rotundine* intervention. Subsequently, 385 DEGs were identified by high-throughput sequencing analysis of *rotundine*-treated cells, of which 85 were up-regulated and 300 were down-regulated. KEGG and GO enrichment analysis of DEGs showed that DEGs were mainly involved in signaling pathways such as the Hippo signaling pathway and intestinal immune network for IgA production, and were enriched in the negative regulation of execution phase of apoptosis, plasma membrane, and receptor antagonist activity biological processes. However, the specific functions and pathways need to be determined by further research. PPI networks play a role in biological cell function and signal transduction [20,21,22,23]. In our study, a potential association between DEGs could be identified by PPI network maps. This result further confirms the DEGs interactions and related pathways.

The prognosis of DEGs in COAD can be observed to further identify the targets of *Rotundine* inhibition in COAD. In the present experiment, we found that *MEF2B*, *CCDC187*, *PSD2, RGS16, PLXDC1, ARV1, HELB*, *ASIC3*, *PLCH2*, *IGF2BP3*, *CLHC1*, *DNHD1*, *SACS*, *H1-4*, *ANKRD36*, and *ZNF117* all had poor and statistically significant prognosis in COAD tissues. Among these 16 prognosis-related DEGs only *ARV1* was lowly expressed in COAD, and the remaining genes were highly expressed. Several research teams have reported aberrant expression of mRNA in COAD. It has been shown that *MEF2B* is significantly up-regulated in ulcerative colitis and is a biomarker for the treatment of ulcerative colitis [24]. *CTB-35F21.1-PSD2* fusion transcripts are highly recurrent in COAD [25]. *RGS16* is highly expressed in COAD tissues and has a low overall survival rate [26]. Several studies have shown that high expression of *PLXDC1* is one of the factors that make the prognosis of COAD patients poor [27,28,29]. The expression of *IGF2BP3* in tissue specimens was found to be a biomarker for the diagnosis of colorectal cancer in endoscopic biopsies [30]. *SACS* is high heterogeneity of expression in COAD and is a predictor of patient survival prognosis [31]. Doxorubicin inhibits the synthesis of *H1-4* through the inhibition of DNA synthesis to achieve the effect of treating colorectal cancer [32]. High expression of *ANKRD36* was strongly associated with poor prognosis of COAD [33]. A nonsense mutation in *ZNF117* predisposes the induction of COAD [34]. These findings are consistent with our findings. It further confirmed the accuracy of our findings. Therefore, we concluded that these 16 genes could be used as effective biomarkers for the treatment of COAD. In addition, we found a positive correlation between these prognosis-related DEGs.

Notably, these prognosis-related DEGs were associated with inositol phosphate metabolism, axon regeneration, inflammatory mediator regulation of TRP channels, systemic lupus erythematosus, apelin signaling pathway, and other cancer-related pathways are closely related. Cellular metabolic activities have a huge impact on cancer development and progression [35,36]. Moreover, the Inositol phosphate metabolism signaling pathway is associated with inositol pyrophosphate. Inositol pyrophosphate, as a metabolic messenger, is positively correlated with cellular energy homeostasis [37]; these dysregulated genes are also involved in a variety of biological processes, including enterobactin transmembrane transporter activity, enterobactin transport, and inner dynein arm. It has been reported that enterobactin has a strong affinity for iron and deprives iron in cells to attenuate the proliferation of cancer [38,39]. Furthermore, enterobactin can be considered an effective anti-cancer agent. Therefore, we believe that *rotundine* inhibits the development of colorectal cancer by regulating the transfer of enterobactin. However, more experiments are needed to verify the detailed regulatory pathways. Overall, our results suggest a potential regulatory role of these 16 prognosis-related DEGs in COAD. However, the specific pathway regulatory mechanism remains to be demonstrated by further experiments. Prognostic modeling combined with an online database revealed that the high-risk group had a poorer prognosis than the low-risk group and was significantly different. In addition, combining clinical data of colorectal cancer patients also revealed that these 16 prognosis-related genes were closely associated with age and clinical analysis of colorectal cancer patients.

In this study, 50 μM, 100 μM, 150 μM, and 200 μM *rotundine* significantly inhibited the growth of colorectal cancer cells in an in vitro study. Bioinformatic analysis was performed and found to be associated with poor prognosis in colorectal cancer patients. *Rotundine* has been extensively studied in animal and human models. Low doses of *rotundine* produced no side effects in mice and relieved pain in patients [40]. 40 mg/kg *rotundine* selectively inhibited OCT2 to reduce renal injury without impairing anti-tumor function [41]. In much Chinese literature, adverse reactions in clinical trials of *rotundine* include nausea, vomiting, dizziness, chest tightness, and weakness. However, adverse reactions were occasionally found and generally did not result in serious consequences. Therefore, more time and experimental trials are needed to study *rotundine* in vivo.

In the present study, we constructed a prognostic model associated with 16 prognosis-related DEGs. The value of these 16 genes in determining the prognosis of COAD was further validated by the model. This model was used to explore the risk factors for the prognosis of colorectal cancer patients. Our results suggest that *MEF2B*, *CCDC187*, *PSD2*, *RGS16*, *PLXDC1*, *ARV1*, *HELB*, *ASIC3*, *PLCH2*, *IGF2BP3*, *CLHC1*, *DNHD1*, *SACS*, *H1-4*, *ANKRD36*, and *ZNF117* can be used as predictors of overall survival prognosis in patients with COAD.

## 5. Conclusions

In conclusion, *rotundine* inhibits proliferation, migration, and invasion of colorectal cancer but does not induce cell apoptosis. High-throughput sequencing and bioinformatics analysis showed that *MEF2B*, *CCDC187*, *PSD2*, *RGS16*, *PLXDC1*, *ARV1*, *HELB*, *ASIC3*, *PLCH2*, *IGF2BP3*, *CLHC1*, *DNHD1*, *SACS*, *H1-4*, *ANKRD36*, and *ZNF117* could be the targets of action of *rotundine* to inhibit colorectal cancer, which may be for further provide new biomarkers for the diagnosis and treatment of colorectal cancer.

## Figures and Tables

**Figure 1 jpm-13-00550-f001:**
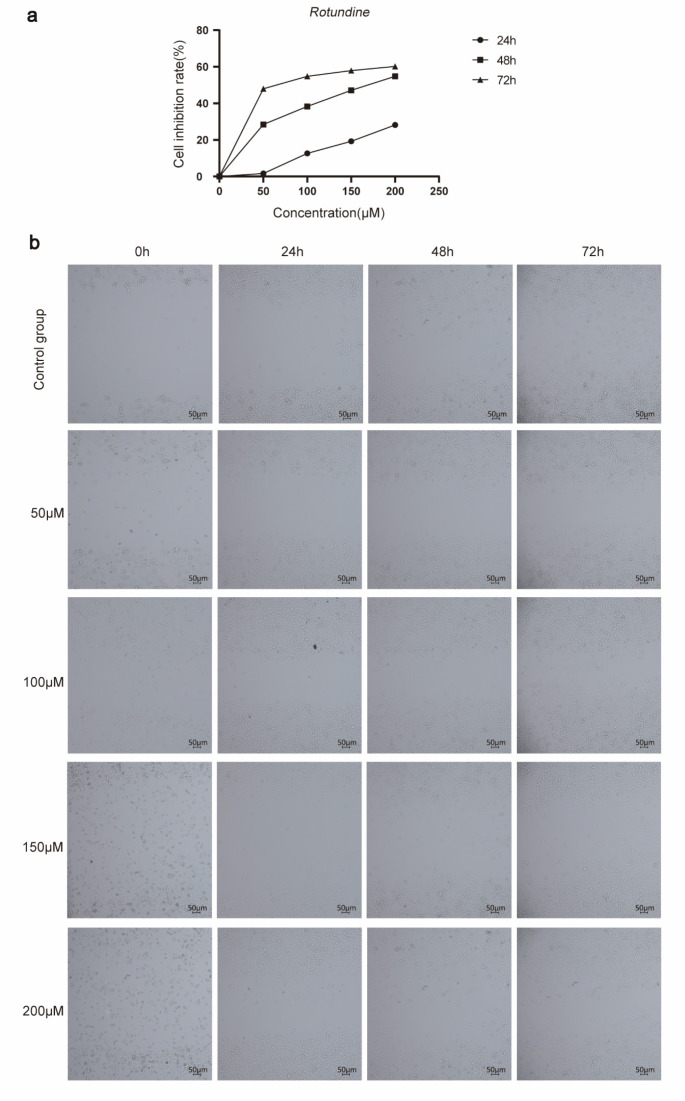
*Rotundine* intervention in SW480 cells experiments. (**a**) MTT assay. (**b**) Cell scratch assay.

**Figure 2 jpm-13-00550-f002:**
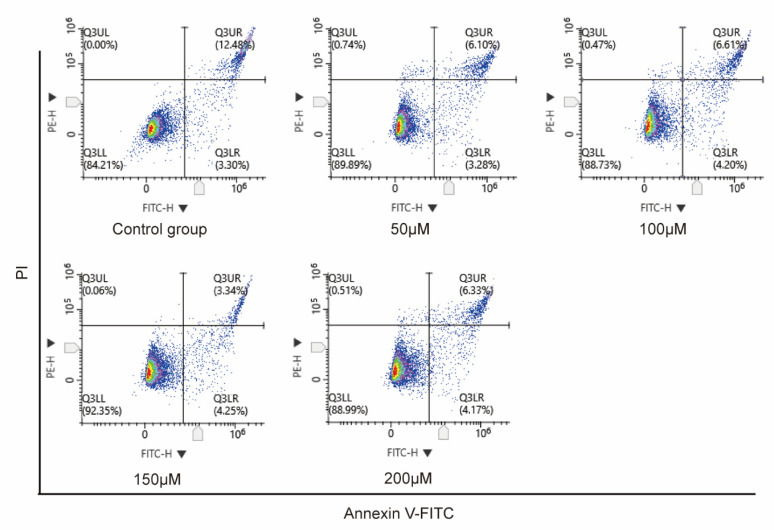
Cell apoptosis. *Rotundine* (0, 50, 100, 150, and 200 μM) intervention in SW480 was followed by annexin V-FITC/PI staining to assess cell apoptosis.

**Figure 3 jpm-13-00550-f003:**
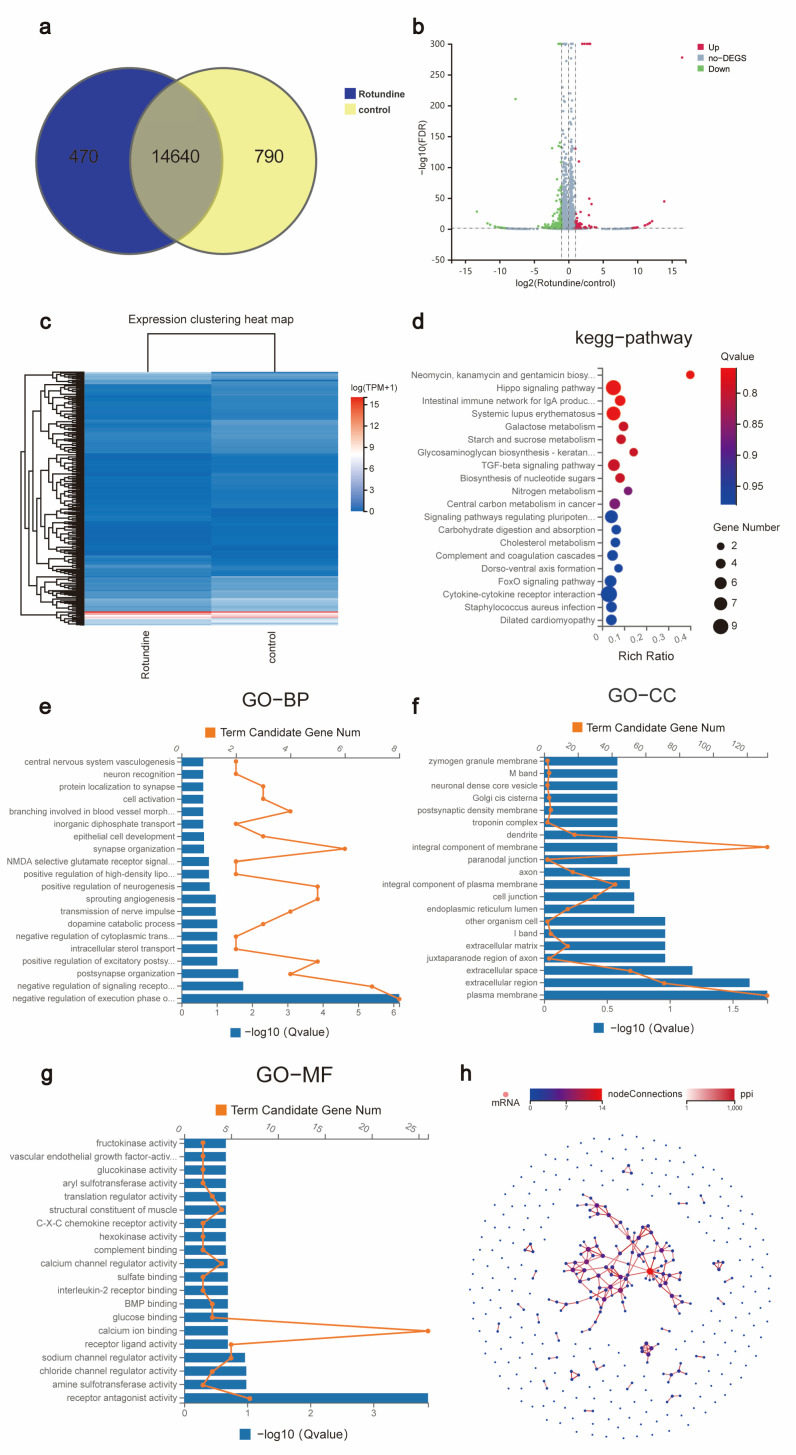
Identification and analysis of DEGs based on high-throughput sequencing. (**a**) High-throughput sequencing Venn diagram. (**b**,**c**) Distribution of DEGs in volcano and heat maps. (**d**) KEGG enrichment analysis. GO enrichment analysis including BP (**e**), CC (**f**), and MF (**g**). (**h**) PPI network.

**Figure 4 jpm-13-00550-f004:**
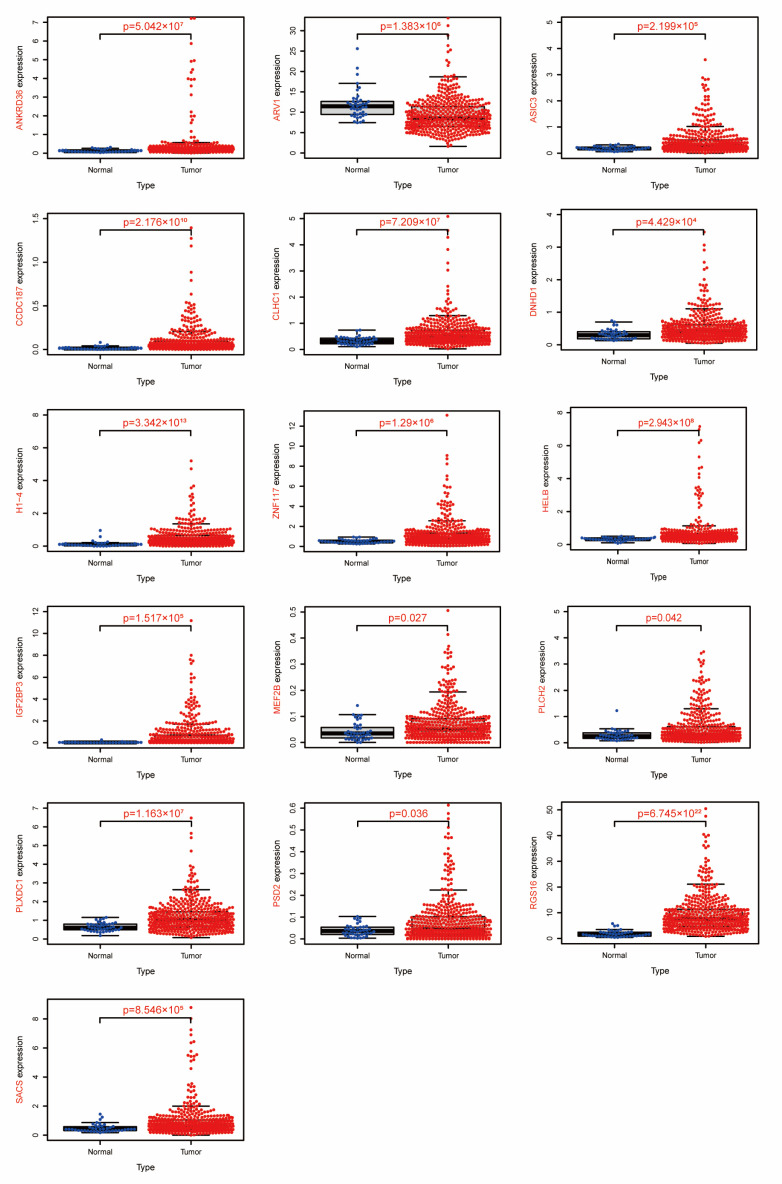
Expression of 16 DEGs in tumor tissues and normal tissues based on the TCGA-COAD cohort. Blue dots represent normal tissue and red represent colorectal cancer tissue.

**Figure 5 jpm-13-00550-f005:**
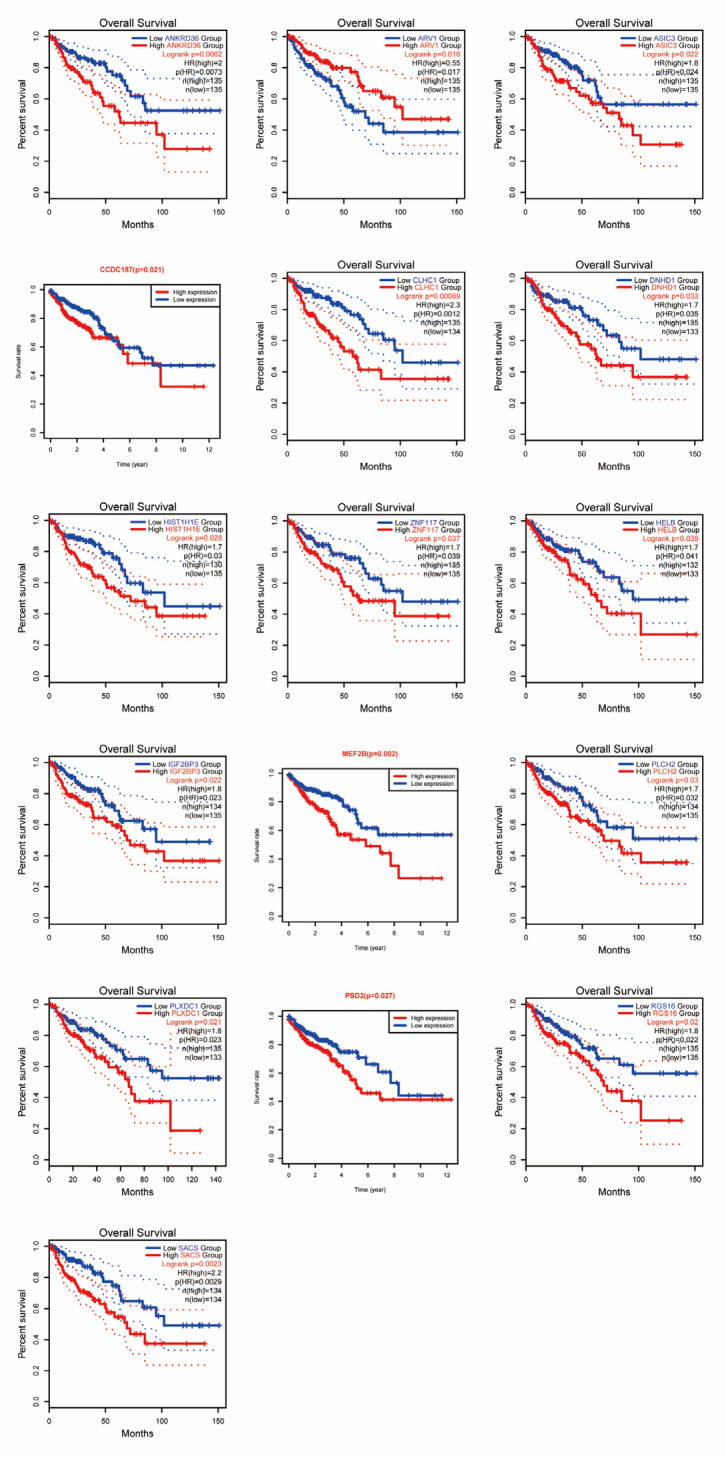
Correlation analysis of 16 DEGs with COAD prognosis.

**Figure 6 jpm-13-00550-f006:**
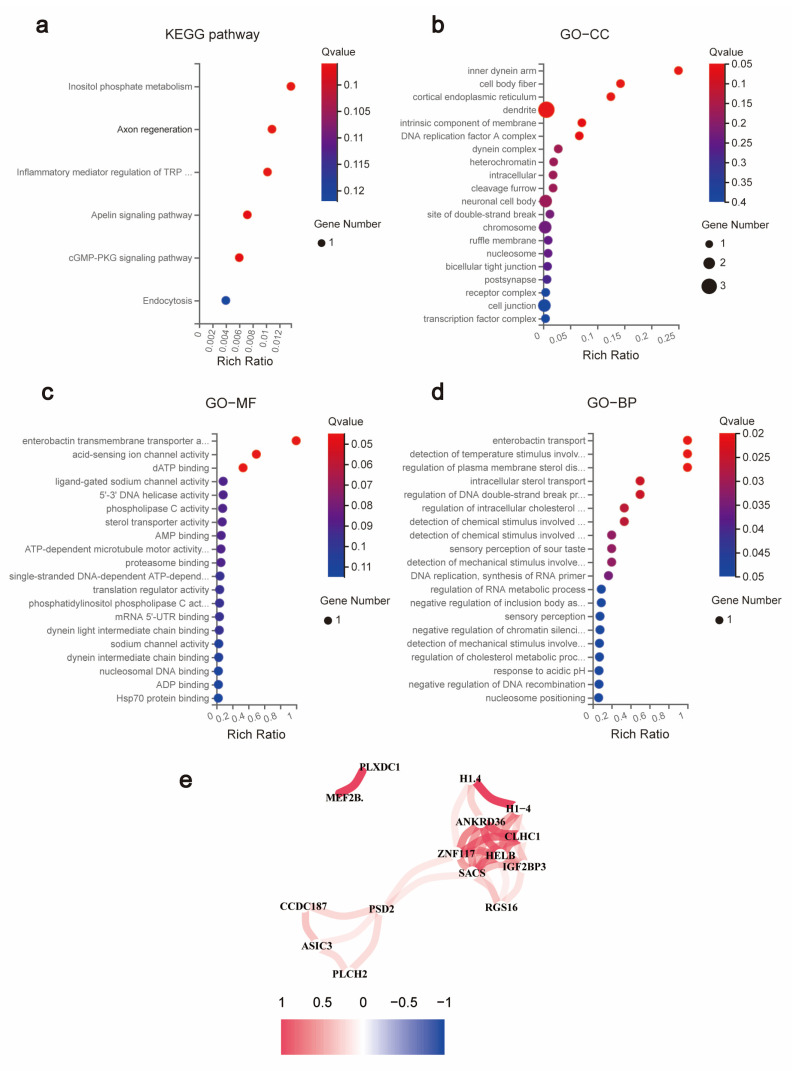
Functional enrichment and correlation analysis of 16 prognosis-related DEGs. (**a**) KEGG pathway analysis. (**b**) GO-CC analysis. (**c**) GO-MF analysis. (**d**) GO-BP analysis. (**e**) Correlation analysis.

**Figure 7 jpm-13-00550-f007:**
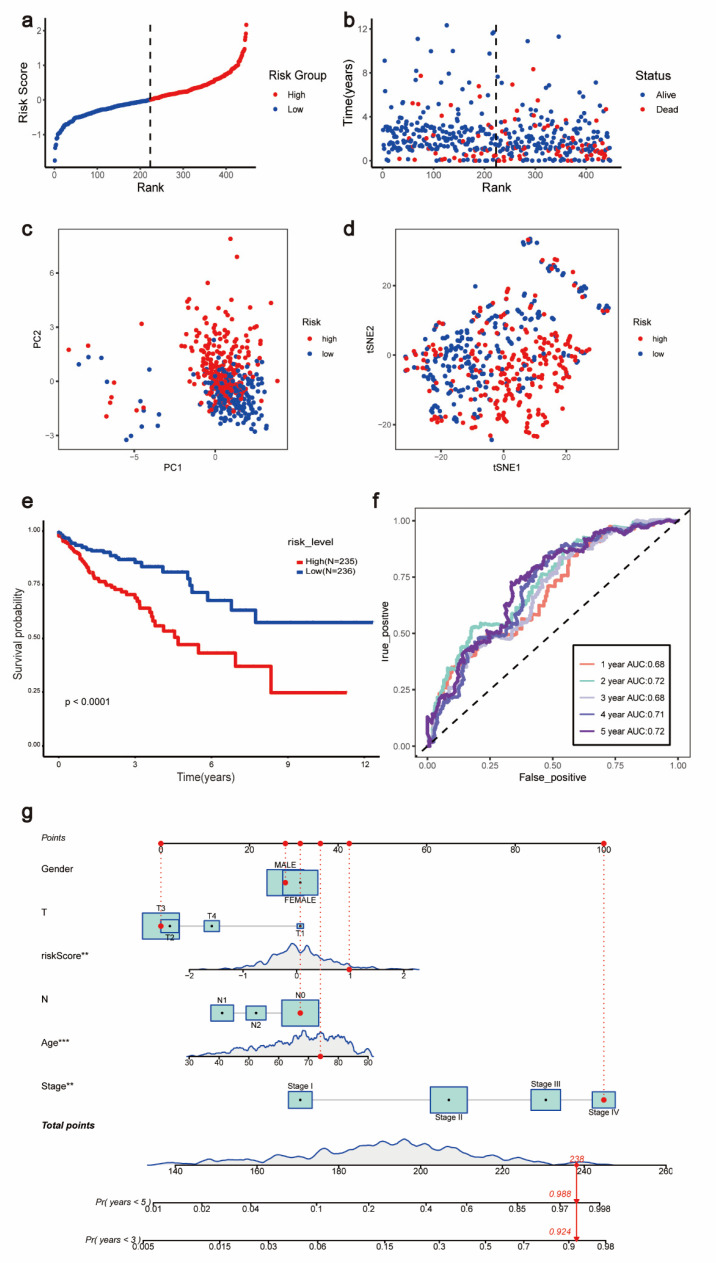
Validation of the prognostic model. The TCGA-COAD cohort was divided into high-risk and low-risk groups based on median risk scores. (**a**) risk curves. (**b**) risk scatter plots. (**c**) PCA. (**d**) t-SNE. (**e**) Kaplan–Meier plotter. (**f**) ROC curves. (**g**) Nomo plots. **, *p* < 0.01; ***, *p* < 0.001.

**Table 1 jpm-13-00550-t001:** The 385 differentially expressed genes (DEGs) obtained by high-throughput sequencing.

385 Differentially Expressed Genes (DEGs) Obtained by High-Throughput Sequencing				
Gene ID	Gene Symbol	Type	log_2_ (*Rotundine*/Control)	FDR (*Rotundine*/Control)
100271849	MEF2B	mRNA	−1.074476505	0.005394141
9651	PLCH2	mRNA	−1.719892081	2.65 × 10^−5^
92797	HELB	mRNA	−1.387023123	0.008073432
9311	ASIC3	mRNA	−1.55359833	4.19 × 10^−7^
84249	PSD2	mRNA	−1.201633861	0.005584584
6004	RGS16	mRNA	−1.332575339	1.51 × 10^−4^
64801	ARV1	mRNA	1.633872101	4.99 × 10^−8^
399693	CCDC187	mRNA	−1.232660757	0.004063028
57125	PLXDC1	mRNA	−1.106915204	0.004030209
10643	IGF2BP3	mRNA	−1.117197143	1.14 × 10^−133^
130162	CLHC1	mRNA	−1.282399731	1.86 × 10^−5^
144132	DNHD1	mRNA	−1.22571278	9.33 × 10^−8^
26278	SACS	mRNA	−1.133362096	8.77 × 10^−63^
3008	H1-4	mRNA	−3.058893689	9.15 × 10^−5^
375248	ANKRD36	mRNA	−1.024627189	5.93 × 10^−11^
51351	ZNF117	mRNA	−13.30092449	7.74 × 10^−29^

## Data Availability

The datasets in the current paper are available from the corresponding author on reasonable request.

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
