# Peer review of "High-Throughput Sequencing Reveals That Rotundine Inhibits Colorectal Cancer by Regulating Prognosis-Related Genes"

_jpm, 2023, doi:10.3390/jpm13030550_

Round 1

Reviewer 1 Report

Huang et al. demonstrated the potential mechanism of rotundine in inhibiting colorectal cancer growth. This study is novel in that it investigates the role of an herb-extracted chemical in cancer treatment. However, the study is immature because multiple claims stay in situ with no proper validations:

1.     Please validate the DEGs with qPCR in your tumor cell samples. They do not always behave the same.

2.     Please indicate the concentration used in the high-throughput sequencing study. Please also discuss whether the concentration is physiologically relevant; namely, whether the concentration the author used are not toxic to normal tissues.

3.     Validation of selected biomarkers is needed, potentially through knock down or knock not or drug inhibition.

4.     Apoptosis or necrosis was not evaluated in the pilot study. Please add corresponding assay.

5.     Along with that, MTT assay is a metabolic activity assay, which does not directly reflect proliferation in this situation. A better assay like EdU/BrdU incorporation is suggested.

6.     It is very interesting that antibiotics biosynthesis was upregulated, which is supposed not to be expressed in mammalian cells. Contamination of mRNA may be involved.

7.     In figure 2h, “a potential association of DEGs was found…..” This does not serve any function in this study. Please discuss why this finding is important.

8.     Again, GO and KEGG are important bioinformatics tools. However, it also needs some form of validations.

9.     In discussion, the author used a bold and irrelevant claim: The effect of Chinese medicine in treating tumors has the advantages of less toxic side effects and better patient tolerance. The statement might be correct (though questionable). However, a chemical extracted and purified from a Chinese herb is not equivalent to Chinese medicine. It is irrelevant to this study.

10.  Introduction is too short and needs a better introduction to roundine.

Reviewer 2 Report

In the manuscript, the authors presented a study of the effects of rotundine on inhibiting colorectal cancer by regulating prognosis-related genes. The idea of the study is sound, and the correlation between bioinformatics studies and the MTT assay was good. The references were new, and the conclusion required improvement to show up the research idea. A few points need to be modified to improve the manuscript, as follows:

1. A list of abbreviations must be added.

2. Introduction: 
1) "According to the Global Cancer Statistics 2020 report," we are now in 2023. So, authors should update the information using the most recent reports. 
2) Fig. 1, (b) Cell scratch assay resolution is poor; please add other images with higher resolution. 
3) All gene names should be written in italic format. 
4) The impact of these results on colorectal cancer patients must appear in the discussion section.

Reviewer 3 Report

Overall, the paper is well put together, highly innovative, with a thorough research protocol.

The introduction could potentially benefit from the following suggestions:

-        I suggest that the authors mention more details about previously published studies about Rotundine and inhibition of proliferation, more exactly when referring to the breast cancer example.

-        The authors should explain the abbreviations used, such as AMPK or MTT.

            The Materials and methods section could be improved by:

-        Elaborating on the Rotundine obtaining process and what kind of standardized solution was used.

-        Providing definitions of all used abbreviations.

            Finally, the Discussion section could be improved by comparing the present findings with in vivo studies.        

Round 2

Reviewer 1 Report

I will give greenlight to all the responses except the one about cell death. The paper cited does not sufficiently support that Rotundine does not affect apoptosis in colorectal cancer cells because the cells used in Xia et al. are MCF-7 cells which are breast cancer cell line. I insist that the author should perform Annexin-V/PI assay or other cell death relevant experiment on the cells used in this study.

Author Response

Response: We appreciate the comments and are very honored to adopt your suggestions. We performed an assay of cell apoptpsis. Roundine (0, 50, 100, 150, 200μM) was applied to SW480 cells for 72h, and cell apoptpsis was analyzed by flow cytometry using Annexin V-FITC /PI staining. The results revealed that Roundine did not induce cell apoptpsis. Combining the results of MTT and cell scratch assay, our results suggested that Roundine can’t induce cell apoptosis, but can inhibits the growth of colorectal cancer cells by suppressing cell proliferation, migration and invasion. We have added and marked this information in the Materials and Methods section on line 100, the Results section on line 157, and the Discussion section on line 275 of the manuscript. Thank you again for suggesting this, it was very helpful for our study.